# RhoA Signaling in Neurodegenerative Diseases

**DOI:** 10.3390/cells11091520

**Published:** 2022-05-01

**Authors:** Sissel Ida Schmidt, Morten Blaabjerg, Kristine Freude, Morten Meyer

**Affiliations:** 1Department of Neurobiology Research, Institute of Molecular Medicine, University of Southern Denmark, DK-5000 Odense, Denmark; sischmidt@health.sdu.dk (S.I.S.); morten.blaabjerg1@rsyd.dk (M.B.); 2Department of Neurology, Odense University Hospital, DK-5000 Odense, Denmark; 3Department of Clinical Research, University of Southern Denmark, DK-5000 Odense, Denmark; 4BRIDGE—Brain Research Inter-Disciplinary Guided Excellence, Department of Clinical Research, University of Southern Denmark, DK-5000 Odense, Denmark; 5Section for Pathobiological Sciences, Department of Veterinary and Animal Sciences, Faculty of Health and Medical Sciences, University of Copenhagen, DK-1870 Frederiksberg, Denmark; kkf@sund.ku.dk

**Keywords:** Ras homolog gene family member A (RhoA), Rho-associated coiled-coil-containing kinase (ROCK), Parkinson’s disease, Alzheimer’s disease, Huntington’s disease, amyotrophic lateral sclerosis

## Abstract

Ras homolog gene family member A (RhoA) is a small GTPase of the Rho family involved in regulating multiple signal transduction pathways that influence a diverse range of cellular functions. RhoA and many of its downstream effector proteins are highly expressed in the nervous system, implying an important role for RhoA signaling in neurons and glial cells. Indeed, emerging evidence points toward a role of aberrant RhoA signaling in neurodegenerative diseases such as Parkinson’s disease, Alzheimer’s disease, Huntington’s disease, and amyotrophic lateral sclerosis. In this review, we summarize the current knowledge of RhoA regulation and downstream cellular functions with an emphasis on the role of RhoA signaling in neurodegenerative diseases and the therapeutic potential of RhoA inhibition in neurodegeneration.

## 1. Introduction

The Rho GTPase family belongs to the RAS superfamily of small GTPases that are guanine nucleotide-dependent molecular switches involved in the regulation of numerous cellular processes including cytoskeletal rearrangement, cell migration, and cell death. In humans, the Rho GTPase family consists of 20 members, the most well-studied members being Ras homolog gene family member A (RhoA), Ras-related C3 botulinum toxin substrate 1 (Rac1), and cell division cycle 42 (Cdc42) [1,2]. Rho GTPases are rapidly activated by various cell surface receptors and cycle between an inactive guanosine diphosphate (GDP)-bound state and an active guanosine triphosphate (GTP)-bound state that is required to initiate downstream signaling cascades through a wide range of effector or target proteins.

Dysregulation of Rho GTPases has been studied in relation to cancer progression for a number of years [3], but accumulating evidence supports a role in traumatic brain injury and neurodegenerative diseases as well [4,5,6,7].

Neurodegenerative diseases are characterized by the progressive loss of specific subsets of neurons. Although the cause of cell death is not fully understood, many of the diseases share an abnormal accumulation of misfolded peptides or proteins in the brain and spinal cord. These include aggregates of α-synuclein in Parkinson’s disease (PD), amyloid-β plaques (Aβ) and inclusions of hyperphosphorylated tau in Alzheimer’s disease (AD), polyglutamine protein aggregates in Huntington’s disease (HD), and inclusions of TAR DNA-binding protein (TDP)-43 in amyotrophic lateral sclerosis (ALS). Other shared pathological mechanisms include defective degradation pathways and protein quality control, dysfunctional mitochondrial homeostasis, autophagy and lysosomal dysregulation, and inflammation [8].

Within the central nervous system, Rho GTPases have been implicated in nearly all steps of brain development [9]. Combined with their widespread cellular activity and alterations in neurodegenerative tissues, targeting Rho GTPases in neurodegenerative diseases may have significant therapeutic potential.

In this article, we review current knowledge on the specific role of RhoA signaling in neurodegenerative diseases and the potential of interfering with this signaling pathway for disease intervention. We start with a thorough description of RhoA regulation from GTP cycling and post-translational modifications to transcriptional control. This is followed by a summary of downstream targets of RhoA relevant to cellular functions involved in neurodegenerative diseases. Finally, we discuss known contributions of this pathway to disease progression in PD, AD, HD, and ALS and the indication for RhoA-inhibiting strategies as a disease-modifying approach in neurodegeneration.

## 2. RhoA Structure and Regulation of RhoA Activity and Expression

The Rho family GTPases show high amino acid sequence and structural homology, comprising a core guanine nucleotide-binding domain (G-domain), an insert domain, and a C-terminal hypervariable domain [10,11].

The G-domain contains five conserved amino acid motifs, G1–G5, that are responsible for GTP and GDP binding and coordination of conformational changes. The G1 motif, also known as the P-loop (phosphate-binding loop), is responsible for binding the β,γ-phosphate of the guanine nucleotide and a Mg^2+^ ion required for nucleotide binding, while the G4 and G5 motifs bind the guanine base. The G2 and G3 motifs are part of the switch I and switch II regions, respectively, which sense whether GDP or GTP is bound and change their conformation accordingly. This conformational change in the switch regions is universal to small GTPases and is known as the “loading-spring” mechanism. Regulatory proteins and effector proteins sense the conformational differences and interact with the switch regions. The insert domain (located between the G4 and G5 motifs) and the C-terminus also play a role in regulatory and effector protein binding. The C-terminal end of Rho GTPases contains a hypervariable domain and a CAAX motif. The hypervariable domain functions as a protein-binding site and has the highest level of sequence variability among the Rho GTPases, accounting for their binding to specific regulatory or effector proteins, while the CAAX motif undergoes post-translational lipid modifications crucial for membrane targeting (Figure 1) [10,11].

Rho GTPases are dependent on GTP binding and the concomitant conformational change that regulates their ability to activate effector proteins and consequent signal transduction. It has become apparent, however, that Rho GTPase regulation exceeds GTP binding and includes a wide range of post-translational modifications (PTMs) to ensure appropriate spatiotemporal activation [11]. Furthermore, Rho GTPases are subjected to regulation at the level of gene expression and to post-transcriptional regulation by microRNAs (miRNA) [12,13].

### 2.1. GDP/GTP Cycling

Similar to most small GTPases, RhoA acts as a molecular switch cycling between an inactive GDP-bound conformation and an active GTP-bound conformation. This conformational transition depends on GDP/GTP exchange processes and GTP hydrolysis, which is mediated by four main regulatory proteins. In response to extracellular stimuli emanating from cell surface receptors, guanine nucleotide exchange factors (GEFs) catalyze the exchange of GDP for GTP, leading to the conformational changes in the switch I and switch II regions required for activation of the small GTPases. Conversely, GTPase-activating proteins (GAPs) enhance the relatively slow intrinsic GTPase activity of the small GTPases, resulting in GTP hydrolysis to GDP and inorganic phosphate (Pi) and inactivation of the GTPases [5,14]. Inactivation is further facilitated by guanine nucleotide dissociation inhibitors (GDIs) that prevent dissociation of GDP from the GTPase and thus retain the inactive GDP-bound GTPases in the cytosol. This dual role of the GDIs is mediated by their N-terminal and C-terminal domains. The N-terminal domain interacts with the switch I and switch II regions of the GTPases, restricting their flexibility and preventing GDP/GTP exchange, while the C-terminal domain contains a hydrophobic pocket that is important in extracting the prenylated GTPases from the membrane [15]. Before the GTPase can be activated by GEFs, it needs to be dissociated from the GDI as GEFs cannot directly act on the GTPase–GDI complex. This step is facilitated by GDI displacement factors (GDFs) (Figure 2A) [16].

Several GEFs and GAPs have been reported to target RhoA. Due to the high amino acid sequence similarity among the Rho GTPases, however, many of the identified GEFs and GAPs show only limited binding specificity, which makes spatiotemporal regulation necessary to allow specific signaling [17]. For example, the first mammalian RhoGEF identified (Dbl), as well as the Dbl-RhoGEF family members Vav1-3, can activate RhoA, but also Rac1 and Cdc42 [10]. Another Dbl-RhoGEF family member, Trio, activates both RhoA and Rac1 but not Cdc42 [18]. The RhoGEFs p115-RhoGEF, p190-RhoGEF, leukemia-associated RhoGEF (LARG), and PSD-95/Disc-large/ZO-1 homology (PDZ)-RhoGEF are more specific as they activate the Rho isoforms RhoA, RhoB, and RhoC but no other Rho family members such as Rac1 and Cdc42 [19]. RhoGEFs that show isoform specificity among RhoA, RhoB, and RhoC are very rare. One example is XPLN, which shows activity toward RhoA and RhoB but not RhoC due to an amino acid difference (Ile43 vs. Val43) at the N-terminus of the Rho isoforms [20,21]. Another example is SmgGDS, which specifically activates RhoA and RhoC but not RhoB [22]. With respect to spatiotemporal regulation, GEFs and GAPs are regulated by protein–protein interactions for localization to specific areas within the cell or by undergoing post-translational modifications. GEF regulation and GAP regulation have been extensively reviewed elsewhere [17,23]. Two examples of GEF regulation by cellular localization are p115-RhoGEF and neuroepithelial transforming gene 1 (Net1). p115-RhoGEF is localized and binds to active heterotrimeric G protein-coupled receptors Gα12 and Gα13 in the plasma membrane, leading to RhoA activation provided that RhoA has been translocated to the plasma membrane near the Gα12 and Gα13 [24]. Net1 is confined to the nucleus, and in response to EGF stimulation, it is acetylated at K93/95, which initiates translocation to the cytosol, where it can access RhoA at the plasma membrane and stimulate its activity [25]. Hence, Net1 is also an example of post-translational regulation of RhoGEFs. Similarly, Vav2 requires post-translational phosphorylation on Tyr174 catalyzed by the Src tyrosine kinase to bind RhoA [26].

RhoGAPs that can differentiate among the Rho isoforms have not yet been identified, but GAPs distinguishing among Rho and other Ras family members have been reported. For example, p190-RhoGAP inactivates RhoA and not Cdc42 and Rac1 [27], but it does not differentiate among RhoA, RhoB, and RhoC [28]. Likewise, the RhoGAP ARHGAP21 shows specificity for RhoA and RhoC but not for Cdc42 [29]. Graf1 is another RhoGAP example that preferentially stimulates the GTPase activity of RhoA and Cdc42 [30]. Interestingly, the RhoA scaffold effector protein Rhotekin binds GTP-bound RhoA and inhibits both the intrinsic and GAP-enhanced GTPase activity of RhoA; this is why it is widely used in pull-down assays for activated RhoA [31].

In contrast to the vast number of known RhoGEFs and RhoGAPs, only three RhoGDIs have been described in mammals [15]. RhoGDI-1 is the most abundantly expressed, and it interacts with several Rho GTPases, including Rac1 and Cdc42, and the Rho isoforms RhoA and RhoC, but not RhoB [32,33]. Likewise, RhoGDI2 has been found to interact with most Rho GTPases, including RhoA, cdc42, and Rac [34,35], while RhoGDI-3 predominantly interacts with RhoB and RhoG [36].

Different agents have been reported to act as GDFs for Rho GTPases [16]. For example, the neurotrophin receptor p75NTR directly binds the RhoGDI-1 and initiates RhoA activation by releasing RhoA from RhoGDI-1 [37]. Interestingly, the p75NTR-RhoGDI-1 complex formation has been found strengthened by the myelin-derived protein Nogo and myelin-associated glycoprotein (MAG) for neurite outgrowth inhibition, while the neurotrophin nerve growth factor (NGF) interferes with the binding between p75NTR and RhoGDI-1 that stimulates neurite outgrowth [37]. Other proteins, such as the Ezrin/radixin/moesin (ERM) proteins, also induce activation of RhoA by acting as RhoGDFs [38]. Similarly, Iκβ kinase γ/Nuclear Factor-κβ-Essential Modulator (IKKγ/NEMO) can function as a RhoGDF as it facilitates the dissociation between RhoA and RhoGDI resulting in RhoA activation [39]. Moreover, different RhoGDI PTMs can function as a GDF event by modulating the affinity of RhoGDIs to specific Rho GTPases [16]. For example, acetylation of RhoGDI-1 on Lys127 and Lys141 was recently shown to reduce its binding to RhoA [40]. Similarly, oxidation of RhoGDI by hydrogen peroxide leads to dissociation of the RhoA-RhoGDI complex [41]. Table 1 lists examples of GEFs, GAPs, GDIs, and GDFs that target RhoA.

### 2.2. Post-Translational Modifications of RhoA

A wide range of different PTMs regulate RhoA. These include prenylation, phosphorylations, and ubiquitinations that are required for regulating RhoA subcellular localization, activation, and inactivation.

#### 2.2.1. Prenylation

Spatiotemporal regulation of Rho GTPase activation is based on the general notion that membrane-bound Rho GTPases are active, while the cytosolic pool is inactive. This is due to membrane binding being necessary for their interaction with membranous GEFs, which then allows GTP-mediated activation and subsequent effector protein interactions [11,46]. Membrane association requires C-terminal prenylation, which is an irreversible PTM; for RhoA, this involves the addition of a geranylgeranyl (20-carbon chain) moiety to the cysteine residue (Cys190) in the CAAX motif. This step is catalyzed by the geranylgeranyltransferase enzyme I (GGTase-I), which transfers the geranylgeranyl moiety from geranylgeranyl pyrophosphate, produced as a product of the mevalonate pathway involved in cholesterol synthesis, to RhoA. The prenylation step takes place in the cytosol and is followed by proteolytic removal of the last three C-terminal (AAX) residues by the protease RAS-converting enzyme 1 (RCE1) and methylation of the prenylated Cys190 by isoprenylcysteine carboxyl methyltransferase (ICMT) at the cytoplasmic surface of the endoplasmic reticulum (ER) [47,48]. Subsequently, RhoA needs to be translocated from the ER membrane to the active site at the plasma membrane. This process requires stabilization by RhoGDI as the absence of membranes will cause the highly hydrophobic geranylgeranyl moiety to impair the ability of RhoA to fold properly in the cytosol, eventually leading to its proteasomal degradation. By inserting the geranylgeranyl moiety into the hydrophobic pocket of the RhoGDI, the moiety is shielded from exposure to water in the cytosol, and GDIs are therefore also said to have chaperone-like functions (Figure 2A,B and Table 2) [15,46].

#### 2.2.2. Phosphorylations

Rho GTPases are highly modified by reversible phosphorylation. Phosphorylation that occurs close to C-terminal lipid modifications can alter the subcellular localization of the Rho GTPase, while phosphorylation of residues within the G-domain affects GTP/GDP cyclin and/or the interaction with effector proteins (Figure 2B and Table 2).

The most well-studied phosphorylation event of RhoA is protein kinase A (PKA) and cyclic GMP-dependent protein kinase (PKG)-mediated phosphorylation on Ser188, close to the prenylated Cys190 in the C-terminus [49,50,56]. This phosphorylation inhibits RhoA activity by favoring interaction with RhoGDI, leading to its extraction from the plasma membrane [50,51,55], and protects GTP-bound RhoA from ubiquitin-mediated proteasomal degradation [52]. This contrasts with the classical model of small GTPase membrane cycling in which the inactive GDP-bound GTPase is the substrate for the GDI-mediated translocation to the cytosol. Therefore, phosphorylation might provide a mechanism to inactivate GTP-bound GTPases by triggering disengagement from their site of action at the membrane, accumulating a reservoir of GTP-loaded GTPases in complex with GDI in the cytosol. Dephosphorylation has thus been suggested as a fast mechanism for mobilizing GTP-bound GTPases back to the membrane for effector protein interaction and signal transduction in the absence of activation by GEFs (Figure 2A) [49]. Furthermore, Ser188 phosphorylation has been found to inhibit RhoA binding to its effector protein Rho-associated coiled-coil-containing kinase (ROCK) but does not affect its binding to other RhoA effector proteins, including rhotekin and mammalian diaphanous (mDia), thereby specifically inhibiting RhoA/ROCK signaling [53,54].

Other kinases, such as Ste20-related kinase (SKL) and AMP-activated protein kinase subunit alpha 1 (AMPKα1), have also been reported to inactivate RhoA by Ser188 phosphorylation [57,58]. Similarly, protein kinase C (PKC) phosphorylates RhoA on Ser188, but also on Thr127 in the insert domain of RhoA, and has been found to form a complex with RhoA in the cytosol in order to facilitate its translocation to the plasma membrane [59].

Another site of phosphorylation-mediated inactivation of RhoA is Ser26. Mammalian Ste20-like kinase 3 (Mst3) phosphorylates RhoA on Ser26 [60]. Adjacent to the Ser26 residue is Lys27 placed in the switch I region, which is known to be a critical residue for interaction between RhoA and its GEFs [77]. The inhibiting effect of the Ser26 phosphorylation might therefore be caused through interference with GEF binding and hence GDP/GTP exchange.

Extracellular signal-related kinase (ERK) phosphorylates RhoA on Ser88 and Thr100, which has been reported to upregulate RhoA activity as evaluated by its binding capacity to the effector protein Rhotekin [61]. On the other hand, ERK phosphorylation of RhoA on an unknown serine residue was found to induce a conformational change exposing the Lys135 acceptor site for ubiquitination, targeting RhoA for proteasomal degradation [62].

In addition to serine and threonine phosphorylation, RhoA is subjected to tyrosine phosphorylation. Within the switch I and switch II regions, the two tyrosine residues Tyr34 and Tyr66 are reported as phosphorylation sites targeted by the tyrosine kinases Bcr-Abl and Src [63]. Phosphorylation of these residues inactivates by preventing effector protein binding [63] and GEF interactions to the switch regions of RhoA [78,79]. In contrast, Src-mediated phosphorylation of Tyr42 has recently been reported to activate RhoA by causing RhoGDI dissociation and GEF-mediated GTP activation followed by effector protein binding [26,64]. However, Tyr42 has also been reported to be phosphorylated by the tyrosine kinase c-Met, which leads to ubiquitin-mediated proteasomal degradation of RhoA [65].

#### 2.2.3. Ubiquitinations

Several Rho GTPases are subjected to ubiquitination that target them for degradation by the proteasome system to regulate protein turnover, but may also regulate their spatiotemporal activities by targeting them to different cellular compartments [80]. Ubiquitination is a multistep process that involves the covalent attachment of ubiquitin to cellular proteins. Briefly, ubiquitin is first activated by an E1 ubiquitin-activating enzyme and is then transferred to an E2 ubiquitin-conjugating enzyme E2. Finally, a mono- or polyubiquitin chain is transferred to a lysine residue on the target protein catalyzed by an E3 ubiquitin ligase [81].

RhoA is ubiquitylated by three different E3 ubiquitin ligase complexes: the SMAD-specific E3 ubiquitin protein ligase 1 (SMURF1), SKP1-CUL1-F-box FBXL19 (SCF^FBXL19^), and the BTB/POZ domain-containing adaptor for CUL3-mediated RhoA degradation (CUL3^BACURD^) ligase complex. While SMURF1 ubiquitinates active GTP-bound RhoA on Lys6 and Lys7 targeting RhoA for proteasomal degradation [66,67], CUL3^BACURD^ ubiquitinates only GDP-bound inactive RhoA [68,69]. In contrast, SCF^FBXL19^ targets both the active and inactive forms of RhoA on Lys135, but the process is dependent on prior Erk2-mediated phosphorylation of RhoA [62]. This difference in RhoA binding might be due to Lys6/7 being localized in close proximity to the first G1 motif in the G-domain responsible for GDP/GTP binding, whereas Lys135 is distant from the GTP/GDP binding sites and resides in the insert domain; this may enable SCF^FBXL19^ to ubiquitinate RhoA in both the GTP- and GDP-bound states (Figure 2B and Table 2).

#### 2.2.4. Additional PTMs

In addition to prenylation, phosphorylation, and ubiquitination, other PTMs of RhoA have been reported such as oxidation, nitration, adenylation, and transglutamination (Figure 2B and Table 2).

Cys16 and Cys20 in the G1 domain of RhoA can be oxidized by reactive oxygen species (ROS), leading to the formation of a disulfide bond between the two residues that prevents guanine nucleotide binding and GEF association, thereby inactivating RhoA [70]. More recently, however, it was reported that Cys16/20 oxidation of RhoA leads to RhoA activation by reducing the affinity of RhoA toward RhoGDI and increasing the association with the RhoA-specific GEF Vav2 [41,71]. This positive effect of Cys16/20 oxidation might also require Tyr42 phosphorylation as P-Tyr42 serves as a binding site for Vav2 [16,26].

Protein nitration in response to NO and superoxide can modify protein structure and function since the addition of a nitro group to tyrosine introduces a negative charge to the modified tyrosine. RhoA nitration on Tyr34 has been reported in response to LPS-induced NO increase, leading to increased RhoA activity. As Tyr34 is located in the switch I region, nitration of this residue possibly increases RhoA activity by inducing a “GEF-like” movement of the switch I region, resulting in faster GDP release and GTP reload [72].

Tyr34 has also been found subjected to adenylation by Fic domain-containing proteins. The addition of adenosine monophosphate (AMP) to the Tyr34 residue results in RhoA inhibition, probably because of steric hindrance of the GDP/GTP binding site in the switch I region [73].

Finally, various types of transglutamination of the Gln63 residue have been found to abolish the intrinsic and GAP-stimulated GTPase activity of RhoA, leading to its permanent activation [74,75,76].

### 2.3. Transcriptional Regulation of RhoA

RhoA is expressed in most tissues at varying levels. Several transcription factors have been shown to regulate *RHOA* expression, including c-Myc, Max, hypoxia-inducible factor 1 alpha (HIF-1α), nuclear factor-κB (NF-κB), and signal transducer and activator of transcription 6 (STAT6), together with several reported miRNAs (Figure 2C) [82,83].

Most of the work on transcriptional regulation of RhoA has been done in cancer models. In breast cancer cells, c-Myc has been found to heterodimerize with the transcription factor Max and to bind to the E-box motif CACGTG in the *RHOA* promoter in cooperation with the coactivator complex Skp2-Miz1-p300, hereby activating *RHOA* transcription [84]. In colon cancer cells, the transcription factors SMAD family member 4 (SMAD4) and specific protein 1 (SP1) activated *RHOA* expression, while c-Myc was reported to interfere with the binding of SP1 to the *RHOA* promoter, leading to downregulation of *RHOA* expression [85]. In neuronal cells, c-Myc cooperated with the co-activator PHD finger protein 8 (PHF8) for *RHOA* expression [86]. Therefore, transcriptional control of *RHOA* by c-Myc might depend on both the coactivators and the cell type. In contrast, HIF-1α transcriptional activation of *RHOA* has consistently been reported across a number of different cell types, including breast cancer cells [87], cardiac myocytes [88], neuroblastoma cells [89], mesenchymal stem cells [90], and chondrocytes [91].

In a study of bronchial smooth muscle cells, the *RHOA* promoter also demonstrated binding to the transcription factors STAT6 and NF-κB [92]. In arterial smooth muscle cells, the nitric oxide (NO)/PKG pathway was found to phosphorylate the activating transcription factor 1 (ATF-1), triggering its binding to the *RHOA* promoter and activate transcription [93]. Moreover, the CCAAT box in the promoter serves as a binding site for various transcription factors, including nuclear factor Y A (NFYA) and B (NFYB), and CCAAT enhances binding protein beta (CEBPB) and delta (CEBPD) [94].

### 2.4. Post-Transcriptional Regulation of RhoA by miRNAs

miRNAs are small noncoding RNAs with a length of 17-24 nucleotides that serve as modulators of gene expression at the post-transcriptional level. They bind to complementary sites on 3′ untranslated regions of target mRNAs to inhibit their translation by ribosomes or induce their degradation [95]. Numerous miRNAs regulate the expression of RhoA [82], including miR-155 [96], miR-31 [97,98], miR-122 [99], and miR-146a [100], but these have mainly been studied in relation to cancers [82,83]. In neuronal cells, miR-133b, miR-185, miR-340, and miR-142 have been found to suppress RhoA expression (Figure 2C).

miR-133b is enriched in midbrain dopaminergic neurons and was found to be deficient in the parkinsonian midbrain [101]. In an MPP^+^-induced cell culture PD model, miR-133b was significantly reduced, while RhoA protein levels were upregulated [102]. Overexpression of miR-133b promoted neurite outgrowth of primary dopaminergic neurons and was able to ameliorate the axon degeneration caused by MPP^+^ by post-transcriptional inhibition of RhoA. Similarly, miR-133b overexpression has been reported to increase neurite outgrowth in PC12 neurons via targeting RhoA expression [103] and to be critical for neural functional recovery after spinal cord injury and stroke [104,105,106].

In mouse models of schizophrenia, miR-185 was downregulated in schizophrenia-associated brain areas, the prefrontal cortex, and the hippocampus [107,108]. The reduced miR-185 expression caused deficits in dendritic complexity and spine development in hippocampal neurons [108], possibly mediated by increased RhoA expression as the two validated miR-185 targets, RhoA and Cdc42 [109], have altered expression levels in schizophrenia [110,111,112]. Another RhoA-targeting miRNA linked to dendrite formation is miR-340, which has been shown to promote dendrite extension through suppressing RhoA expression in melanocytes [113]. Furthermore, miR-142 promoted neurogenic differentiation of adipose-derived stem cells by targeting RhoA [114,115].

## 3. Downstream Targets of RhoA

Active GTP-bound RhoA transduces upstream signals from various cell surface receptors, including integrins, tyrosine kinase receptors, G protein-coupled receptors, and cytokine receptors, by interacting with downstream effector molecules.

The effector proteins for RhoA described so far fall primarily into two classes: scaffold proteins such as rhophilin, rhotekin, kinectin, and mammalian diaphanous-related formin (mDia) and serine/threonine protein kinases such as protein kinase N (PKN), citron kinase, and ROCK [116]. RhoA and many of its effector proteins are highly expressed in the nervous system, implying an important role for RhoA signaling in neurons and glial cells [1,117]. Indeed, RhoA regulates a diverse set of cellular processes such as cytoskeleton modulation, cell death, mitochondrial homeostasis, autophagy, inflammation, and gene transcription.

### 3.1. Cytoskeletal Dynamics

A major function of Rho GTPases is to regulate the assembly and organization of the actin–myosin cytoskeleton that is important for neuronal growth cone dynamics, dendritic spine formation, and axonal extension and guidance. Many neuronal studies support the general model that Rac and Cdc42 stimulate these processes, while RhoA inhibits them and causes axonal and dendritic retraction and spine and synapse loss. Hence, RhoA and Rac/Cdc42 typically exhibit an antagonistic relationship to regulate neuronal morphology [1,7].

In most cell types, the effect of RhoA on cytoskeletal dynamics is primarily mediated by its activation of ROCK, which via phosphorylation and activation of LIM kinase (LIMK) increases the phosphorylation and inactivation of cofilin 1 (an actin-depolymerizing and severing factor) to stabilize actin filaments. This leads to reduced actin turnover and inhibition of neurite outgrowth [118]. ROCK also phosphorylates the myosin light chain (MLC) subunit of the actin-based motor protein myosin IIb, leading to actomyosin assembly and contractility. This occurs both through a direct interaction and indirectly by ROCK-mediated inhibition of myosin light chain phosphatase (MLCP) through phosphorylation of its myosin phosphatase-targeting subunit 1 (MYPT1) [7]. Furthermore, several neuron-specific ROCK targets have been identified. Collapsing response mediator protein 2 (CRMP-2) is a microtubule-binding protein that stimulates axon growth by promoting microtubule assembly. ROCK-mediated phosphorylation of CRMP-2 impairs its ability to bind tubulin and induces growth cone collapse [119]. MAP2, tau, and neurofilament are also neuron-specific substrates for ROCK, and their microtubule polymerizing activity and neurofilament assembly are inhibited by ROCK-mediated phosphorylation, leading to inhibition of neurite elongation and shortened neurites [120].

mDia is another prominent RhoA effector protein in relation to cytoskeletal regulation. mDia is a formin molecule that catalyzes actin nucleation and polymerization through its interaction with the actin-binding protein profilin [121]. As mDia activation leads to the formation of actin filaments and ROCK activates myosin to crosslink them, ROCK and mDia cooperate for actomyosin assembly and contractility [122].Collectively, these actions of RhoA signaling drive neurite retraction and loss of spines and synapses (Figure 3).

RhoA-mediated regulation of cytoskeletal dynamics also plays an essential role in cytokinesis. Increased RhoA signaling has been associated with tumorigenesis [3] but is also important to CNS development. Active RhoA promotes the formation and stabilization of the cleavage furrow through its control of actin polymerization and drives the contraction of the contractile ring through its stimulation of MLC phosphorylation. The effector proteins involved are mDia, ROCK, and citron kinase. While mDia regulates actin polymerization and formation of the contractile ring, ROCK phosphorylates MLC and MYPT1 to activate myosin for furrow ingression, and citron K is required for late cytokinesis stages of abscission [123]. Similarly to ROCK, citron kinase phosphorylates MLC, but it di-phosphorylates it at Thr18 and Ser19 and causes a different localization of myosin in the furrow, particularly at late stages of cytokinesis [123]. Citron kinase also regulates proper spindle orientation by promoting nucleation and stability of astral microtubules. Whether citron kinase is a downstream effector protein of RhoA or merely an upstream regulator has been debated as citron kinase has also been found to regulate the activity of RhoA, most likely through a scaffolding function [124]. Another RhoA scaffolding protein, anillin, plays a fundamental role in the assembly and stabilization of the contractile ring by interacting with RhoA, actin filaments, microtubules, septins, and membrane phospholipids [125] (Figure 3).

While citron kinase is ubiquitously expressed in proliferating cells, the isoform citron N, which is also a scaffold protein for RhoA, is expressed only in postmitotic and fully differentiated neurons [124]. Citron N is enriched in dendritic spines and is associated with the endomembrane system, in particular the Golgi complex, where it maintains Golgi architecture by modulating local actin polymerization together with ROCK [126,127]. A novel RhoA effector protein FAM65A is also associated with the Golgi complex. Activated RhoA binds FAM65A, which in turn binds cerebral cavernous malformation-3 protein (CCM3) and the interacting mammalian STE20-like protein kinase (MST). Upon binding to RhoA, the FAM65A/CCM3/MST complex relocates away from the Golgi complex to the cytoplasmic membrane, relieving the inhibitory effect of MST4 on Golgi reorientation. This is essential for both membrane trafficking from the Golgi and the orientation of the Golgi toward the leading edge of the cell [128].

In summary, RhoA signaling regulates many fundamental cellular processes that rely on the cytoskeletal organization (Figure 3).

### 3.2. Cell Death

Similar to their opposing effects on cytoskeletal dynamics, Rac typically promotes neuronal survival, while RhoA induces neuronal apoptosis [1].

The antiapoptotic effects of Rac are antagonized by RhoA/ROCK signaling through a number of downstream ROCK targets involved in both mitochondrial-dependent intrinsic apoptotic pathways and death receptor-dependent extrinsic apoptotic pathways. ROCK can phosphorylate and activate phosphatase and tensin homologue (PTEN), which is a negative regulator of the phosphatidylinositol 3-kinase (PI3K)/Akt pathway initiating apoptosis through released Akt suppression of proapoptotic Bcl-2 family proteins [129].

The cytoskeletal rearrangement induced by ROCK activation can also be involved in apoptosis. Actomyosin contractility stimulated by ROCK-mediated MLC phosphorylation can regulate morphological apoptotic events occurring during the execution phases of apoptosis, including cell contraction, plasma membrane blebbing, nuclear disintegration, and fragmentation of apoptotic cells; it can also trigger the initiating phases of apoptosis through regulation of death receptor assembly or loss of cell adhesion [130].

Furthermore, RhoA can induce apoptosis via direct activation of the mitogen-activated protein kinases (MAPK) p38 and c-Jun N-terminal kinase (JNK) signaling pathways, which induce apoptosis through activation of proapoptotic Bcl-2 family members, e.g., Bad and Bax [1] (Figure 3).

### 3.3. Mitochondria

In addition to the well-established role of RhoA signaling in neurite retraction, spine and synapse loss, and neuronal apoptosis, recent evidence points toward RhoA-mediated alterations of mitochondrial function and homeostasis. Mitochondria are dynamic organelles that frequently move, fuse, and divide to meet the metabolic needs of the cell. The dynamic balance between fusion and fission of mitochondria is vital to ensure mitochondrial health, and increasing evidence has indicated that mitochondrial fission participates in the initiation of mitochondria-dependent apoptosis [131]. Activated ROCK was found to promote mitochondrial fission and subsequent apoptosis in dopaminergic neurons via the activation of dynamin-related protein 1 (Drp1) [132]. Whether the ROCK-mediated activation of Drp1 is a phosphorylation or dephosphorylation event is still not clear. It seems that ROCK can directly phosphorylate Drp1 to increase the fission activity of Drp1 in cardiomyocytes and podocytes [133,134], while in dopaminergic neurons, a ROCK-mediated dephosphorylation of Drp1 was reported to induce mitochondrial fission [132]. Interestingly, the downstream ROCK target calcineurin can dephosphorylate Drp1 and cause mitochondrial fission [135,136]. Hence, ROCK might indirectly dephosphorylate Drp1 through calcineurin activation, but whether this is the case or ROCK directly phosphorylates Drp1 for activation might be dependent on the cell type.

RhoA signaling is also involved in mitochondrial movement. The cellular distribution of mitochondria is controlled both by their translocation along the microtubule system and by their retention at specific metabolic active sites within the cell [137]. Through its downstream effector mDia, RhoA inhibits mitochondrial movement and ensures the anchoring of mitochondria to the actin cytoskeleton [138].

Furthermore, it was recently suggested that ROCK inhibits Parkin-mediated mitophagy via PTEN antagonization of the PI3K/Akt pathway [139] as Akt-mediated activation of hexokinase 2 (HK2) is required for its translocation to damaged mitochondria and subsequent recruitment of Parkin to initiate mitophagy [139,140] (Figure 3).

### 3.4. Autophagy

After the ubiquitin–proteasome system (UPS), autophagy is the most important intracellular degradation system for dysfunctional proteins and organelles [141]. Several studies have shown that inhibition of ROCK or direct downregulation of ROCK expression results in the activation of both autophagy and the UPS, indicating that RhoA/ROCK signaling negatively regulates the degradation pathways [142,143,144,145,146]. The exact downstream targets of RhoA/ROCK responsible for this effect are still not fully elucidated, but it has been suggested that ROCK regulates autophagosome formation via PI3K and the JNK/Bcl-2/Beclin 1/Vps34 pathway [145]. In a study of A53T α-syn degradation, inhibition of ROCK in SH-SY5Y cells increased autophagy via activation of JNK and subsequent phosphorylation of Bcl-2. When phosphorylated, Bcl-2 dissociates from Beclin 1, allowing Beclin 1 to complex with Vps34 and initiate autophagosome formation [145]. Another possible connection between ROCK and autophagy could be its effect on Akt signaling as Akt regulates mTOR-dependent autophagy [147].

The biogenesis and trafficking of autophagosomes are highly dependent on cytoskeletal dynamics, including actin assembly, membrane–cytoskeletal scaffolds, and actomyosin-based transport [148]. Therefore, it is also very likely that the effect of RhoA on autophagy is indirectly mediated by its regulation of cytoskeletal dynamics. For example, RhoA/ROCK signaling was shown to regulate intracellular redistribution of lysosomes [149,150], suggesting that RhoA/ROCK signaling could indeed regulate the trafficking from early autophagosomes to late autolysosomes. More studies are needed to investigate the connection between RhoA/ROCK and autophagy in more detail.

### 3.5. Neuroinflammation

One of the mechanisms leading to neuronal cell death in neurodegenerative diseases is the activation of microglia [151]. Increasing evidence indicates that RhoA/ROCK signaling has an important role in microglial activation by regulating microglial migration, phagocytosis, and release of inflammatory factors such as ROS, tumor necrosis factor α (TNFα), interleukin-6 (IL-6), IL-1β, and NO [152,153]. The role of RhoA/ROCK signaling in regulating cytoskeletal dynamics, as described above, is particularly important for the activation of microglial migration and phagocytosis [154]. Interestingly, ROCK inhibition has been reported to shift microglia from their proinflammatory M1 phenotype toward the anti-inflammatory M2 phenotype by inhibiting NF-кB activity and expression of the proinflammatory cytokines IL-1β, IL-6, and TNFα, while increasing the expression of the anti-inflammatory cytokine IL-10 [155,156,157]. ROCK activates the proinflammatory transcription factor NF-кB through phosphorylation of the IкB kinase subunit β (IKKβ). Activated IKKβ then phosphorylates the inhibitor of NF-кB (IкB), leading to its degradation and release of NF-кB that translocates to the nucleus and activates transcription of proinflammatory factors [39].

RhoA regulates ROS production in microglial cells through activation of NADPH oxidase (NOX) [158]. The mechanism involves ROCK-mediated phosphorylation and activation of the p47^PHOX^ subunit of NOX [159]. Furthermore, as mentioned in the RhoA PTM section, factors related to ROS generation also directly influence the activity of RhoA. Hydrogen peroxide has been reported to both induce oxidation of RhoA and RhoGDI, abolishing their complex formation [41], and increase activation of the RhoGEF Vav2 through Src hyperoxidation [26], both leading to increased RhoA activity. A vicious loop of ROS production has recently been suggested in which ROS activates RhoA/ROCK, leading to p47^PHOX^ phosphorylation and NOX activation, thus further increasing ROS levels [159] (Figure 3).

In addition to the essential role of microglia in RhoA-mediated neuroinflammation, other glial cells may also contribute. In astrocytes, RhoA/ROCK signaling plays a major role in determining astrocyte morphology and reactive state [118,160,161], and ROCK inhibition reduces reactive gliosis and increases astrocytic expression of prosurvival genes [162,163]. In oligodendrocytes, RhoA/ROCK signaling causes a failure in oligodendrocyte precursor cell differentiation and process extension required for myelination [164,165] and also regulates ROS production [166].

### 3.6. Gene Transcription

RhoA signaling can regulate the activity of several transcription factors through both actin-dependent and actin-independent pathways. The transcription factor serum response factor (SRF) regulates the serum response element (SRE) found in many promoters, including genes that encode cytoskeletal components such as actin and myosin light chain. SRF requires the coactivator megakaryoblastic leukemia-1 (MKL-1) (also known as myocardin-related transcription factor (MRTF) and megakaryocytic acute leukemia (MAL)). RhoA-mediated actin polymerization through ROCK/LIMK or mDia reduces the interaction between actin and MLK-1, thus inhibiting the actin-dependent nuclear export of MKL-1, which results in its nuclear accumulation and enhanced SRF-mediated transcription [167].

RhoA signaling can mediate actin-independent gene transcription such as the ROCK-mediated activation of the proinflammatory transcription factor NF-кB through phosphorylation of the IкB kinase subunit β (IKKβ), as described above [39]. In addition, the JNK and p38 MAPK cascades result in increased expression of proapoptotic proteins and decreased expression of antiapoptotic proteins through a variety of transcription factors, e.g., activator protein-1 (AP-1) and p53 [168].

Other transcription factors regulated by RhoA include peroxisome proliferator-activated receptor γ (PPARγ), β-catenin, signal transducer and activator of transcription-3 (STAT3), and hypoxia inducible factor-1α (HIF-1α). However, these have mainly been studied in relation to cancers [82].

## 4. RhoA Signaling in Neurodegenerative Diseases

Neurodegenerative diseases such as PD, AD, HD, and ALS are rapidly increasing in incidence worldwide. While common disease mechanisms such as dysfunctional protein clearance, protein aggregation, axonal degeneration, and altered immune response have been characterized, no efficient disease-modifying therapies have so far been developed.

In the following section, we discuss the growing body of literature on the involvement of RhoA signaling in the pathogenesis of PD, AD, ALS, and HD and the effect of inhibiting RhoA signaling in these diseases. Current available strategies to inhibit RhoA signaling, either directly targeting RhoA or targeting its downstream effectors, have been extensively reviewed elsewhere [117]. Table 3 summarizes the studies discussed in this section and lists their RhoA inhibitory strategies.

### 4.1. Parkinson’s Disease

PD is the most common neurodegenerative movement disorder and is caused by the progressive degeneration of nigrostriatal dopaminergic neurons of the midbrain. The cardinal motor symptoms are bradykinesia, resting tremor, and muscular rigidity. As the neurodegeneration develops, other brain regions are also affected, including the meso- and neocortical regions, leading to cognitive and behavioral symptoms [215,216]. The neuropathological hallmark of the disease is the presence of Lewy bodies (ubiquitylated intraneuronal protein inclusions enriched in aggregated α-synuclein) in residual neurons [217]. Sporadic PD accounts for most cases, and its etiology is still unclear. However, studies of gene mutations linked to familial PD have identified a number of mechanisms involved in the dopaminergic neuronal degeneration in PD. These include mitochondrial dysfunction and increased oxidative stress, together with protein misfolding and impairments in the ubiquitin–proteasome and autophagy–lysosomal systems, and neuroinflammation [218,219,220].

The role of RhoA signaling in PD pathogenesis is supported by studies of mice treated with the dopaminergic neurotoxin MPTP that show an upregulation of RhoA and ROCK in the substantia nigra pars compacta [173]. ROCK inhibition was found to protect against the MPTP-induced dopaminergic cell death in both mice and primary cultures of dopaminergic neurons [172,173]. The neuroprotective effect was attributed to inhibition of the MPTP-induced microglial inflammatory response [171,173,174] and to a direct protective effect on the dopaminergic neuritic network in vitro and the striatal axonal innervation in vivo [172,174]. In line with these observations, we recently showed that RhoA signaling is increased in human-induced pluripotent stem cell-derived neurons with mutation in the PD-associated gene *PARK2*. Increased RhoA activity was found to alter migration and impair neuritogenesis, which could be rescued by Rhosin-mediated RhoA inhibition [178]. In primary rat hippocampal and dopaminergic neuronal cultures treated with the neurotoxic pesticide rotenone, RhoA activity was increased and associated with reduced neurite outgrowth, which was normalized by ROCK inhibition using Y27632 [169]. Similarly, rotenone treatment of primary mouse mesencephalic cultures increased RhoA activity, and RhoA inhibition using either C3 transferase or Simvastatin protected the dopaminergic neurons against rotenone-induced neurite damage [177]. As neuritic and axonal degeneration is one of the earliest pathological features of PD [221], inhibition of RhoA signaling might be a promising strategy for early disease intervention.

RhoA inhibition has also been found to reduce α-synuclein expression in MN9D dopaminergic cells. This effect was mediated through the reduction in the nuclear transcription factor SRF, in its coactivator MKL-1, and in GATA-2, a transcriptional regulator of α-synuclein [170]. Similarly, a study of primary dopaminergic neurons and PC12 cells showed that inhibition of RhoA by miR-133b expression attenuated MPTP-induced α-synuclein upregulation and ameliorated axon degeneration [102]. In vivo, ROCK inhibition using Fasudil was shown to decrease midbrain α-synuclein pathology and improve motor and cognitive functions in a transgenic mouse model expressing human α-synuclein with the A53T PD mutation [175]. Fasudil-mediated ROCK inhibition also induced α-synuclein clearance in A53T-overexpressing SH-SY5Y cells in vitro; the underlying mechanism was found to be ROCK inhibition-mediated activation of autophagy via the JNK/Bcl-2/Beclin 1/Vps34 pathway [145]. With respect to microglial activation and neuroinflammation in PD, extracellular α-synuclein has been found to induce microglial ROS production through CD11b integrin-mediated NOX activation via RhoA [176].

ROCK inhibition has also been linked to mitochondrial improvements. In MPTP-treated PC12 cells and a MPTP mouse model, ROCK inhibition using Y27632 rescued Drp1-mediated aberrant mitochondrial fission and apoptosis of dopaminergic neurons [132]. Furthermore, it has been suggested that ROCK inhibitors may be neuroprotective by upregulating parkin-mediated mitophagy. A recent high-throughput screen of ~3000 compounds in HEK293 cells identified several ROCK inhibitors that increased the recruitment of parkin to damaged mitochondria, leading to increased targeting of mitochondria to lysosomes and removal of damaged mitochondria from the cells. The inhibitor SR3677 was the most efficient, and its neuroprotective effect was verified in vivo in a fly PD model subjected to paraquat, a parkinsonian toxin that induces mitochondrial damage [139].

These data suggest that RhoA/ROCK inhibition might be a potential therapeutic strategy for PD as it interferes with both neuritic and axonal phenotypes, mitochondrial and autophagic impairments, α-synuclein pathology, and neuroinflammation. RhoA/ROCK signaling has very recently also been linked to L-DOPA-induced dyskinesia in a rat model of Parkinson’s disease. Although dopamine replacement therapy using the precursor L-DOPA is the main treatment for PD, abnormal involuntary movements known as dyskinesias are one of the most severe complications of chronic L-DOPA treatment. In 6-hydroxydopamine lesioned rats with dyskinesia, RhoA and ROCK were increased in the substantia nigra and striatum, and ROCK inhibition using Fasudil was found to reduce the development of L-DOPA-induced dyskinesia and to inhibit already established dyskinesia affecting the therapeutic effect of L-DOPA [179]. Targeting RhoA/ROCK in PD might therefore be a possible therapeutic approach for more advanced stages of the disease.

### 4.2. Alzheimer’s Disease

AD is the most common form of dementia and is caused by extensive neuronal loss in the hippocampus and cerebral cortex, leading to the typical AD symptoms of memory loss, spatiotemporal disorientation, and behavioral changes. The neurodegenerative process is associated with aberrant accumulation of extracellular Aβ, accumulation of hyperphosphorylated tau in intracellular neurofibrillary tangles (NFTs), and dendritic spine and synapse loss [222].

Accumulating evidence supports a role for RhoA signaling in Aβ aggregation, tau phosphorylation, neuroinflammation, and synaptic damage in AD [223]. Overall RhoA levels were reduced in human AD brains, while remaining RhoA colocalized with hyperphosphorylated tau in NFTs [185]. Moreover, in Aβ precursor protein (APP)-overexpressing transgenic mouse brains, RhoA expression was decreased within synapses but increased in degenerating neurites [185], suggesting that RhoA mislocation is related to neurodegeneration. Exposing human neuroblastoma cells to Aβ was found to increase RhoA activation and subsequent inhibitory CHRM-2 phosphorylation, leading to dystrophic changes in neurite morphology that were reversed by ROCK inhibition using Y27632 [184]. Similarly, soluble Aβ was reported to disrupt actin and microtubule dynamics via activation of RhoA and inhibition of histone deacetylase 6 in cultured hippocampal neurons [189]. Aβ activates RhoA by binding to the p75 neurotrophin receptor (p75^NTR^), and inhibition of this receptor or RhoA directly prevents the deleterious effects of Aβ in hippocampal neurons [186,196].

The opposite relation between Aβ and RhoA also exists, as Aβ production by the secretase-dependent cleavage of APP has been found enhanced by RhoA/ROCK activation [180]. It was recently reported that ROCK directly interacts with APP and phosphorylates its Ser655 residue, promoting amyloidogenic processing of APP by increasing the interaction of β-site APP cleaving enzyme 1 (BACE1) with APP [194]. This suggests that RhoA/ROCK signaling regulates APP processing as well. In line with this, a study found that selective inhibition of the ROCK2 isoform, the most abundantly expressed ROCK isoform in the brain, suppressed BACE1 activity and thereby reduced Aβ levels in cellular and animal models of AD. The mechanism was found to involve altered BACE1 endocytic distribution and promotion of APP traffic to lysosomes [188]. It was later found that also targeting the ROCK1 isoform reduces Aβ levels in primary neuronal cultures by enhancing APP protein degradation, probably through induction of autophagy [191]. This is suggestive of a positive feedback loop where increased RhoA/ROCK activation promotes increased Aβ production, which in turn potentiates RhoA/ROCK activation and enhances the Aβ build-up [191]. Targeting RhoA/ROCK signaling might therefore be a potential strategy to curb Aβ production in AD.

Another pathological hallmark of AD is the formation of neurofibrillary tangles (NFTs) from hyperphosphorylated tau. Tau is a substrate for phosphorylation by ROCK [181], and ROCK inhibition was found to mitigate tau pathology both in vitro and in vivo by reducing total tau levels, tau phosphorylation, and oligomerization as well as facilitating tau degradation through autophagy and proteasome pathways [146,190]. RhoA inhibition using pitavastatin has also been reported to reduce total tau and phosphorylated tau levels in a cellular model of tauopathy and in primary neuronal cultures [187]. Indeed, several studies have demonstrated that statin-mediated RhoA/ROCK inhibition reduces NFT formation and decreases Aβ production [182,183,187,224].

RhoA/ROCK signaling has also been linked to the retraction of dendritic spines in AD. Synapse and dendritic spine loss is one of the main hallmarks of the early phases of AD and is directly correlated with the cognitive decline associated with AD [225]. Oligomeric Aβ was found to induce synaptic loss in neurons via Pyk2-mediated inactivation of the RhoGAP Graf1, leading to RhoA activation and subsequent dendritic spine retraction [193]. Several studies have shown that inhibition of RhoA/ROCK can rescue dendritic spine loss in different AD models. In a D-galactose and aluminum-induced AD-like rat model, paeonol was found to attenuate neuronal dendritic spine loss through inhibition of the RhoA/ROCK/LIMK1/cofilin1 pathway [195]. Inhibition of p75^NTR^-mediated RhoA activation with the TAT-Pep5 peptide or Y-27632-mediated ROCK inhibition protected against oligomeric Aβ-induced spine loss in hippocampal neurons [196]. ROCK inhibition in an APP/presenilin-1 (PS1) transgenic mouse model showed improved learning and memory impairment, accompanied by reduced tau phosphorylation, expression of Aβ and BACE1, and increased expression of synapse-associated proteins and neurotrophic factors in the hippocampus and cortex area of the brain [192].

With respect to microglial activation and neuroinflammation in AD, increased RhoA expression has been reported in reactive microglia in APP/PS1 transgenic and fibrillary Aβ injected mouse models [153]. RhoA/ROCK signaling was found to be essential for Aβ-induced chemotaxis, cytotoxicity, and inflammatory response of BV2 microglia, and ROCK inhibition by Fasudil or Y27632 could efficiently suppress these inflammatory responses [153].

In view of these data, the inhibition of RhoA signaling to combat Aβ and tau-mediated pathologies in AD holds great potential.

### 4.3. Huntington’s Disease

HD is a progressive and fatal autosomal-dominant neurodegenerative disease characterized by a severe loss of GABA’ergic medium spiny neurons in the striatum as well as cortical and hippocampal neurodegeneration, resulting in involuntary body movements, cognitive decline, psychiatric impairments, and eventually premature death. The pathology of HD is caused by a mutation in the huntingtin gene (*HTT*), which results in the expansion of a CAG nucleotide repeat that leads to an elongated polyglutamine tract in the N-terminus of the huntingtin protein (Htt). This expansion destabilizes Htt, leading to misfolding and toxic gain-of-function aggregation of Htt [226].

Several studies have highlighted an important role for RhoA signaling in the etiology of HD. In blood samples and frontal cortex of postmortem brain tissue from HD patients as well as in the striatum of the R6/2 HD mouse model, mRNA expression levels of RhoA, ROCK, and a number of downstream cytoskeletal-related effector proteins were upregulated [207]. From an in vitro screen for molecules that reduce polyglutamine toxicity, the ROCK-inhibitor Y27632 was identified and subsequently shown to reduce Htt aggregation in cultured cells and Htt-mediated neurodegeneration in drosophila [197,198]. The mechanism was later found to involve Y27632-mediated increase in both proteasomal degradation of Htt and macroautophagy [200]. In the R6/2 HD mouse model, Y27632-mediated ROCK inhibition improved rotarod performance and reduced soluble Htt, but it had no effect on Htt aggregation, cellular atrophy in the striatum, or survival, possibly due to suboptimal dosing as no significant reduction of ROCK effector protein phosphorylation was seen [199]. However, intravitreal application of the more potent ROCK inhibitor, Fasudil, was able to rescue the progressive retinopathy in the same HD mouse model [202].

Targeting p75^NTR^ upstream of RhoA signaling has proven beneficial in HD. p75^NTR^ was increased in HD mouse models and in postmortem samples from HD patients [203,204]. Increased p75^NTR^ levels were accompanied by long-term memory deficits, altered LTP, reduced levels of synaptic proteins, decreased dendritic spine density, and increased activation of RhoA [203]. Normalizing p75^NTR^ signaling prevented memory and synaptic deficits in HD mutant mice [203,204,205,208]. Likewise, direct inhibition of RhoA using C3 transferase ameliorated dendritic spine abnormalities in primary hippocampal cultures [203].

In the 3-nitropropionic acid (3-NP) HD rat model, inhibition of RhoA/ROCK signaling using Fasudil or simvastatin inhibited 3-NP-induced neurotoxicity and, interestingly, also mitochondrial dysfunction through a mechanism involving the ROCK/p-Akt/eNOS pathway [206].

Increased activation of RhoA signaling has also been implicated in the dopamine-related vulnerability of striatal medium spiny neurons in HD. Despite ubiquitous expression of mutant Htt throughout the HD brain, it is the GABA’ergic medium spiny neurons in the striatum that predominantly degenerate. A generally accepted hypothesis is that the dopaminergic inputs to these neurons participate in this vulnerability. Dopamine D2 receptor stimulation by dopamine has been found to act in synergy with mutant Htt to increase aggregate formation, neuritic retraction, and striatal death through activation of the RhoA/ROCK/cofilin pathway, which could be reversed by ROCK inhibition [201,209].

These data demonstrate that inhibition of RhoA signaling has a disease-modifying potential in HD. The underlying mechanisms need further investigation, however.

### 4.4. Amyotrophic Lateral Sclerosis

ALS is a fatal neuromuscular disorder characterized by the degeneration of motor neurons of the motor cortex, brain stem, and upper and lower spinal cord. This results in severe weakness of skeletal muscles, causing atrophy and eventually death due to paralysis of the respiratory muscles. While the majority of ALS cases are considered sporadic, most cases of familial ALS arise from genetic mutations in the genes encoding superoxide dismutase 1 (SOD1), Alsin (ALS2), TDP-43, chromosome 9 open reading frame 72 (C9ORF72), and fused in sarcoma (FUS) [227].

Although the pathogenesis of ALS is not fully understood, it is probably multifactorial and includes protein aggregation, excitotoxicity, oxidative stress, mitochondrial dysfunction, disturbances in RNA metabolism, and impaired axonal transport [227]. Activation of the RhoA/ROCK pathway has also been implicated in the pathogenesis of ALS, although studied to a lesser degree and focusing on the disease-modifying effects of ROCK inhibition. ROCK expression was increased in the SOD1^G93A^ transgenic mouse model of ALS and in sporadic ALS patients [210,212]. ROCK inhibition has been shown to preserve neuromuscular junctions and extend the survival of SOD1^G93A^ mice via reduced microgliosis and decreased release of proinflammatory cytokines and chemokines such as TNFα and IL6 [163]. ROCK inhibition also slowed disease progression and reduced motor neuron loss in SOD1^G93A^ mice through reduced ROCK-mediated activation of PTEN and restored activation of the prosurvival kinase Akt [211]. These studies applied Fasudil-mediated ROCK inhibition at the presymptomatic disease stage. ROCK inhibition in a more advanced disease stage in SOD1^G93A^ mice did not show increased motor neuron survival nor reduced microglial infiltration at the neuropathological level, although motor function was improved in male mice [213]. Given the advanced pathology at first symptom onset in the SOD1^G93A^ mouse model of ALS, the ability to beneficially modify motor function is still significant. Interestingly, ROCK inhibition using Y-27632 instead was found to improve axonal regeneration of injured motor axons after sciatic crush in both presymptomatic and symptomatic SOD1^G93A^ mice [214].

## 5. Conclusions

Although neurodegenerative diseases are multifactorial, dysregulated activity of Rho family GTPases has emerged as a possible common feature underlying the degeneration process. This review focused on the specific role of RhoA signaling as this is involved in cytoskeleton modulation and cell death and has recently been linked to neurodegeneration-relevant cellular processes such as mitochondrial homeostasis, autophagy, and neuroinflammation. Future research should further investigate the signaling pathways that are aberrantly activated or disrupted downstream of RhoA to elucidate the promising disease-modifying potential of targeting RhoA signaling in neurogenerative diseases.

## Figures and Tables

**Figure 1 cells-11-01520-f001:**
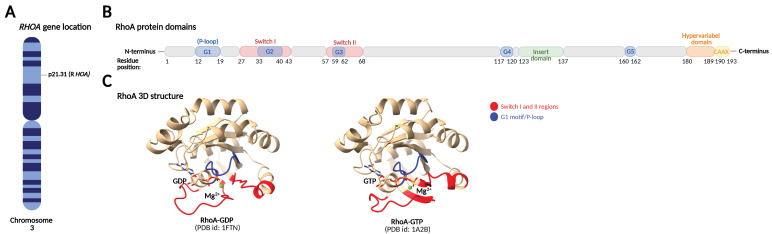
RhoA protein domains and 3D structure. (**A**) The *RHOA* gene is located on the short arm of chromosome 3 (3p21.3). (**B**) RhoA is a monomeric protein comprised of 193 amino acids and a molecular mass of 21.7 kDa. The different domains and their amino acid positions are marked. The G-domain responsible for nucleotide binding contains five motifs, G1–G5, where the G2 and G3 motifs are located in the switch I and II regions that change conformation in response to GTP/GDP binding. The insert domain and the hypervariable C-terminus are involved in regulatory or effector protein binding, while the CAAX motif in the C-terminal end undergoes post-translational lipid modification crucial for membrane targeting. (**C**) 3D structure and conformation of the RhoA–GDP complex and the RhoA–GTP complex. Marked in blue is the P-loop (G1 motif) that binds the β,γ-phosphate of the guanine nucleotide and a Mg^2+^ ion. Marked in red are the switch I and II regions responsible for the conformational change of the protein in response to GTP/GDP binding.

**Figure 2 cells-11-01520-f002:**
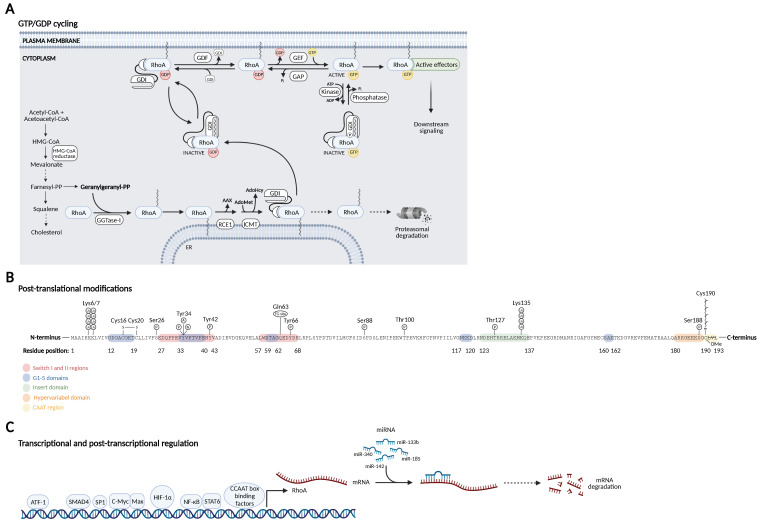
RhoA regulation. (**A**) RhoA cycle between an active GTP-bound conformation at the plasma membrane and an inactive GDP-bound conformation in the cytosol facilitated by prenylation and different regulatory proteins: guanine nucleotide exchange factors (GEFs), GTPase-activating proteins (GAPs), guanine nucleotide dissociation inhibitors (GDIs), and GDI displacement factors (GDFs).(**B**) Post-translational modification events mapped onto the amino acid sequence of RhoA with highlighted locations of the functional protein domains from Figure 1. (**C**) Transcriptional and post-transcriptional regulation of RhoA showing known transcription factors and miRNAs regulating *RHOA* transcription.

**Figure 3 cells-11-01520-f003:**
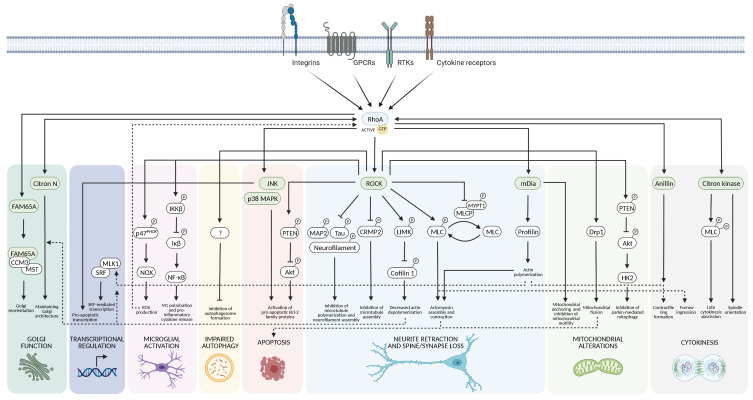
RhoA signaling. RhoA transmits signals from a variety of different cell surface receptors, such as integrins, tyrosine kinase receptors, G protein-coupled receptors, and cytokine receptors, by interacting with downstream effector molecules involved in many cellular processes. These include, Golgi function, gene transcription, inflammation, autophagy, cell death, cytoskeletal modulation, mitochondrial homeostasis, and cytokinesis.

**Table 1 cells-11-01520-t001:** Examples of GEFs, GAPs, GDIs, and GDFs targeting RhoA.

GTP/GDP Regulatory Proteins	Factors	References
GEFs	Dbl	[42,43]
	Vav1-3	[44,45]
	Trio	[18]
	p115-RhoGEF	[19]
	p190-RhoGEF	[19]
	LARG	[19]
	PDZ-RhoGEF	[19]
	XPLN	[20,21]
	SmgGDS	[22]
	Net1	[25]
GAPs	p190-RhoGAP	[27]
	ARHGAP21	[29]
	Graf1	[30]
GDIs	RhoGDI-1	[32,33]
	RhoGDI-2	[34]
GDFs	p75NTR	[37]
	ERM proteins	[38]
	IKKγ/NEMO	[39]

Abbreviations: ERM, Ezrin/radixin/moesin; GAP, GTPase-activating protein; GDI, guanine nucleotide dissociation inhibitor; GDF, GDI displacement factor; GEF, guanine nucleotide exchange factor; IKKγ/NEMO, Iκβ kinase γ/Nuclear Factor-κβ-Essential Modulator; LARG, leukemia-associated RhoGEF; Net1, neuroepithelial transforming gene 1; PDZ-RhoGEF, PSD-95/Disc-large/ZO-1 homology RhoGEF.

**Table 2 cells-11-01520-t002:** Post-translational modifications of RhoA.

PTM	Enzyme/Factor	Sites	Effect	References
Prenylation	GGTase-I	Cys190	Membrane anchoring	[46]
Phosphorylations	PKA	Ser188	Plasma membrane retraction by increasing interaction with GDI	[49,50,51]
			Protects GTP-bound RhoA from ubiquitin-mediated proteasomal degradation	[52]
			Decreases binding to RhoA effector protein ROCK	[53,54]
	PKG	Ser188	Translocation to the cytosol by increasing interaction with GDI	[50,55]
			Protects GTP-bound RhoA from proteasomal degradation	[52,56]
	AMPKa1	Ser188	Inactivation	[57]
	SLK	Ser188	Inactivation	[58]
	PKC	Thr127 Ser188	Translocation to the plasma membrane	[59]
	Mst3	Ser26	Inactivation by hindering GEF interaction	[60]
	ERK	Ser88 Thr100	Upregulates RhoA activity	[61]
		Unknown	Induce ubiquitin-mediated proteasomal degradation	[62]
	Bcr-Abl Src	Tyr34 Tyr66	Inhibits effector protein binding and GEF interactions	[63]
	Src	Tyr42	Activation by GDI dissociation and GEF interaction	[26,64]
	c-Met	Tyr42	Induce ubiquitin-mediated proteasomal degradation	[65]
Ubiquitinations	SMURF1	Lys6 Lys7	Targets active GTP-bound RhoA for proteasomal degradation	[66,67]
	CUL3^BACURD^	Unknown	Targets inactive GDP-bound RhoA for proteasomal degradation	[68,69]
	SCF^FBXL19^	Lys135	Targets both active and inactive RhoA for proteasomal degradation	[62]
Oxidation	ROS	Cys16 Cys20	Inactivation by preventing guanine nucleotide binding and GEF association	[70]
			Activation by GDI dissociation and GEF interaction possible requiring combined P-Tyr42	[41,71]
Nitration	NO	Tyr34	Activation	[72]
Adenylation	Fic domain-containing proteins	Tyr34	Inactivation by steric hindrance of the GDP/GTP binding site in the switch I region	[73]
Transglutamination	Transglutaminase	Gln63	Constitutive activation by abolishing the intrinsic and GAP-stimulated GTPase activity	[74,75,76]

Abbreviations: AMPKa1, AMP-activated protein kinase subunit alpha 1; CUL3^BACURD^, BTB/POZ domain-containing adaptor for CUL3-mediated RhoA degradation ligase complex; ERK, extracellular signal-related kinase; GAP, GTPase-activating protein; GDI, guanine nucleotide dissociation inhibitor; GEF, guanine nucleotide exchange factor; Mst3, mammalian Ste20-like kinase 3; PKA, protein kinase A; PKC, protein kinase C; PKG, cyclic GMP-dependent protein kinase; ROS, reactive oxygen species; SCF^FBXL19^, SKP1-CUL1-F-box FBXL19 ligase complex; SLK, Ste20-related kinase; SMURF1, SMAD-specific E3 ubiquitin protein ligase 1.

**Table 3 cells-11-01520-t003:** Studies linking RhoA signaling to Parkinson’s disease, Alzheimer’s disease, Huntington’s disease, and amyotrophic lateral sclerosis.

Neurodegenerative Disease	Study Model	Finding	Target	Inhibitor(s)	References
Parkinson’s disease (PD)	Rat primary hippocampal and mesencephalic cultures	Rotenone treatment increases RhoA activity and inhibition of ROCK rescues rotenone-induced inhibition of neurite outgrowth	ROCK	Y27632	[169]
	Mouse MN9D dopaminergic cell line	RhoA inhibition leads to neurite extension and reduces α-synuclein expression	RhoA	C3 transferase and db-cAMP	[170]
	MPTP mouse model	ROCK inhibition prevents microglia from eliminating dopaminergic neurons in MPTP-treated mice	ROCK	HA-1077	[171]
	MPTP mouse model and rat primary midbrain dopaminergic neurons	ROCK inhibition enhances survival of dopaminergic neurons and attenuates axonal loss	ROCK	Fasudil	[172]
	MPTP mouse model and rat primary mesencephalic cultures	Upregulation of RhoA and ROCK in the substantia nigra pars compacta of MPTP-treated mice. ROCK inhibition protects against MPTP-induced dopaminergic cell death both in vivo and in vitro	ROCK	Y27632	[173]
	Rat primary mesencephalic cultures	Inhibition of microglial ROCK is essential to protect against MPTP-induced dopaminergic cell death, but ROCK inhibition also induces a direct effect against axonal retraction in surviving dopaminergic neurons	ROCK	Y27632	[174]
	Transgenic mouse model expressing human A53T α-synuclein	ROCK inhibition decreases midbrain α-synuclein pathology and improves motor and cognitive functions	ROCK	Fasudil	[175]
	Rat primary mesencephalic cultures and PC12 cells	MPTP treatment upregulates RhoA expression, and RhoA inhibition attenuates MPTP-induced α-synuclein upregulation and ameliorates axon degeneration	RhoA	miR-133b overexpression	[102]
	Human SH-SY5Y cells	Increased ROCK activity in A53T α-synuclein-overexpressing cells. ROCK inhibition induces clearance of A53T α-synuclein by activating autophagy	ROCK	Fasudil	[145]
	Murine primary microglial cultures	α-synuclein induces microglial ROS production through CD11b integrin-mediated RhoA/NOX activation	RhoA	siRNA	[176]
	Murine primary mesencephalic cultures	Rotenone induces RhoA activation, and RhoA inhibition protects dopaminergic neurons against rotenone-induced neurite damage	RhoA	C3 transferase and simvastatin	[177]
	MPTP-treated PC12 cells and MPTP mouse model	ROCK inhibition rescues Drp1-mediated aberrant mitochondrial fission and apoptosis of dopaminergic neurons both in vitro and in vivo	ROCK	Y27632	[132]
	Human-induced pluripotent stem cell-derived neurons with *PARK2* mutation	Increased RhoA signaling caused altered migration and impaired neuritogenesis, which could be rescued by RhoA inhibition	RhoA	Rhosin	[178]
	Human HEK293 cells, SH-SY5Y cells, and paraquat Drosophila model	A screen of ~3000 compounds identified several ROCK inhibitors that rescued mitochondrial damage by upregulating parkin-mediated mitophagy. The ROCK inhibitor SR3677 was found to be most efficient	ROCK	SR3677	[139]
	6-OHDA lesioned rats	Increased RhoA and ROCK expression in 6-OHDA lesioned rats with dyskinesia. ROCK inhibition reduces the development of dyskinesia and also inhibits already established dyskinesia	ROCK	Fasudil	[179]
Alzheimer’s disease (AD)	Human SH-SY5Y cells	Aβ production by secretase-dependent cleavage of APP is reduced by RhoA/ROCK inhibition	RhoA and ROCK	NSAIDs, C3 transferase, Y27632 and overexpression of dominant negative RhoA	[180]
	Tau-transfected COS7 cells	ROCK phosphorylates tau and reduces the activity of tau to promote microtubule assembly	-	-	[181]
	Mouse N2a cells	Statins stimulate sAPPα shedding by ROCK inhibition	RhoA	Atorvastatin and simvastatin	[182]
	Mouse N2a cells	Statins reduce Aβ production through inhibition of Rho and Rab family proteins	RhoA	Simvastatin and lovastatin	[183]
	Human SH-SY5Y cells	Aβ exposure leads to inhibition of neurite outgrowth through increased RhoA activation, which is rescued by ROCK inhibition	ROCK	Y27632	[184]
	Postmortem human AD brains and APP-overexpressing mouse model	RhoA is decreased in human AD brains, and remaining RhoA colocalizes with hyperphosphorylated tau. In APP-overexpressing mice, RhoA was decreased within synapses but increased in degenerating neurites	-	-	[185]
	Murine primary hippocampal neurons and PC12 cells	Aβ activates RhoA by binding p75NTR. Inhibition of RhoA prevents the deleterious effect of Aβ on cultured hippocampal neurons	RhoA	C3 transferase and overexpression of dominant negative RhoA	[186]
	Human M1C cells expressing tau and murine primary cortical neurons	Inhibition of RhoA/ROCK signaling using pitavastatin reduces total tau and phosphorylated tau levels	RhoA	Pitavastatin	[187]
	Human SH-SY5Y cells, murine primary cortical neurons, HEK293 cells, and 5XFAD mouse model	Selective ROCK2 inhibition reduces Aβ production by inhibiting BACE1 activity. The mechanism involves altered BACE1 endocytic distribution and APP trafficking to lysosomes	ROCK2	SR3677	[188]
	Murine primary hippocampal neurons	Soluble Aβ disrupts actin and microtubule dynamics via activation of RhoA and inhibition of histone deacetylase 6, which is rescued by ROCK inhibition	ROCK	Y27632	[189]
	Human SH-SY5Y cells, murine primary cortical neurons, and tau-expressing drosophila	ROCK inhibition diminishes total and phosphorylated tau levels through enhancing autophagy and reducing tau mRNA	ROCK	shRNA, SR3677, and Fasudil	[190]
	Postmortem human AD brains, murine primary cortical neurons, and ROCK1^−/−^ mice	ROCK1 protein level is increased in human AD brains, and ROCK1 inhibition reduces Aβ levels	ROCK1	shRNA	[191]
	APP/PS1 mouse model	ROCK inhibition attenuated Aβ burden, tau phosphorylation and BACE expression and increased expression of synapse-associated proteins and neurotrophic factors	ROCK	FSD-C10	[192]
	APP/PS1 mouse model, WT mice injected with fAβ and BV2 microglial cells	Increased RhoA expression in reactive microglia in vivo. RhoA/ROCK signaling is essential for Aβ-induced chemotactic migration, cytotoxicity, and inflammatory responses in microglial BV2 cells. ROCK inhibition suppresses the inflammatory responses	ROCK	Fasudil and Y27632	[153]
	Murine primary hippocampal neurons, HEK293 cells, APP mouse model, and Pyk2^−/−^ mouse model	Aβ induces an increase in actin contractility via Pyk2/RhoGAP Graf1/RhoA-regulated ROCK activation, culminating in dendritic spine retraction. Spine loss is rescued by RhoA and ROCK inhibition	RhoA and ROCK	Y27632 and overexpression of dominant negative RhoA	[193]
	APP/PS1 mouse model	APP is a substrate for ROCK, which phosphorylates its Ser655 residue to promote amyloidogenic processing of APP by BACE1. ROCK inhibition rescues Aβ pathology and improves learning and memory in APP/PS1 mice	ROCK	shRNA and Y27632	[194]
	Human M1C cells expressing WT tau, murine primary neurons, and rTG4510 mouse model	ROCK inhibition reduces total tau levels, tau phosphorylation, and oligomerization and upregulates autophagy and proteasome pathways	ROCK	H1152, Y-27632, and Fasudil	[146]
	D-galactose and aluminum rat model	Paeonol rescues neuronal dendritic spine loss through inhibition of the RhoA/ROCK/LIMK1/cofilin1 pathway	RhoA	Paeonol	[195]
	p75NTR^−/−^ murine primary hippocampal neurons	Aβ activates RhoA through p75NTR. Inhibition of p75NTR-mediated RhoA activation or ROCK protects neurons from Aβ-induced dendritic spine pathology	p75NTR and ROCK	TAT-Pep5 peptide and Y27632	[196]
Huntington’s disease (HD)	COS7 cells, HEK293 cells, C17-2 cells, and HD drosophila model	Y27632 was identified in a compound screen to reduce polyglutamine toxicity. Verification in cellular and drosophila models of HD shows reduced Htt aggregation and toxicity	ROCK	Y27632	[197]
	HEK293 cells and rat primary cortical neurons	ROCK inhibition reduces Htt aggregation	ROCK	Y27632	[198]
	R6/2 mouse model	ROCK inhibition improves rotarod performance and reduces soluble mutant Htt in R6/2 mice, but no effect on Htt aggregation, cellular atrophy in the striatum, or lifespan	ROCK	Y27632	[199]
	Mouse N2a cells and MEFs	ROCK inhibition reduces Htt aggregation via activation of the ubiquitin proteasome system and macroautophagy	ROCK	Y27632	[200]
	Mouse primary striatal neurons	Dopaminergic D2 receptor stimulation act in synergy with mutant Htt to increase aggregates formation and striatal cell death through activation of RhoA/ROCK. This could be rescued by ROCK inhibition	ROCK	siRNA, Y27632 and Fasudil	[201]
	R6/2 mouse model	ROCK inhibition improved retinal function in the R6/2 mouse model	ROCK	Fasudil	[202]
	R6/1 and HdhQ7/Q111 knock-in mouse models, p75NTR-overexpressing primary hippocampal neurons and postmortem human HD brains	p75NTR levels is increased in HD mouse models and in post-mortem brain tissue from HD patients. Normalizing p75NTR levels prevented memory and synaptic deficits in HD mutant mice. Inhibition of RhoA normalized dendritic spine density in primary hippocampal cultures	p75NTR and RhoA	shRNA and C3 transferase	[203]
	Mutant BACHD mice	Plasticity in indirect pathway spiny projection neurons from BACHD mutant mice can be rescued by inhibition of p75NTR/RhoA signaling	p75NTR and ROCK	TAT-Pep5 peptide and Y27632	[204]
	R6/1 mouse model	Inhibition of p75NTR rescues dendritic spine loss through negative regulation of RhoA	p75NTR	FTY720	[205]
	3-NP rat model	Inhibition of RhoA/ROCK signaling inhibited 3-NP-induced neurotoxicity and mitochondrial dysfunction	RhoA and ROCK	Simvastatin and Fasudil	[206]
	HD human blood leukocytes, postmortem HD brain tissue, and R6/2 mouse model	Increased mRNA expression of RhoA, ROCK, and downstream cytoskeletal-related effector proteins in leukocytes and frontal cortex of postmortem brain tissue from HD patients and in striatum of R6/2 mice	-	-	[207]
	R6/2 and BACHD mouse models	Normalizing p75NTR signaling reduces key HD neuropathologies, including Htt aggregation and dendritic spine loss, and improves cognition and motor performance in HD mouse models. The effect was mediated by downregulation of ROCK and PTEN	p75NTR	LM11A-31 (ligand)	[208]
	HEK293 cells and mouse primary striatal neurons	The dopamine D2 receptor short isoform, but not the long isoform, is coupled to the RhoA/ROCK/cofilin pathway and its involvement in striatal vulnerability to mutant Htt	-	-	[209]
Amyotrophic lateral sclerosis (ALS)	SOD1-G93A mouse model	Increased ROCK expression in the spinal cord of G93A SOD1 mice	-	-	[210]
	SOD1-G93A mouse model	ROCK inhibition delays disease onset and extends survival in SOD1-G93A mice. ROCK inhibition reduced the phosphorylation of PTEN, resulting in neuronal protection via increasing phosphorylated Akt	ROCK	Fasudil	[211]
	Human ALS skeletal muscle biopsies	Increased ROCK expression in skeletal muscles from sporadic ALS patients	-	-	[212]
	SOD1-G93A mouse model	ROCK inhibition delays onset and extends survival in SOD1-G93A mice via reduced microgliosis and decreased release of proinflammatory cytokines and chemokines	ROCK	Fasudil	[163]
	SOD1-G93A mouse model	ROCK inhibition in a more advanced disease stage in SOD1-G93A mice improved motor function in male mice but did not increase motor neuron survival or reduce microglial infiltration	ROCK	Fasudil	[213]
	SOD1-G93A mouse model	ROCK inhibition improved axonal regeneration of injured motor axons after sciatic crush in SOD1-G93A mice	ROCK	Y27632	[214]

Abbreviations: 3-NP, 3-Nitropropionic acid; 6-OHDA, 6-hydroxydopamine; Aβ, amyloid β; APP, Aβ precursor protein; BACE1, β-site APP cleaving enzyme 1; Drp1, dynamin-related protein 1; Htt, huntingtin; MPTP, 1-methyl-4-phenyl-1,2,3,6-tetrahydropyridine; PS1, presenilin-1.

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
