# Peer review of "RhoA Signaling in Neurodegenerative Diseases"

_cells, 2022, doi:10.3390/cells11091520_

Round 1
Reviewer 1 Report
Title: RhoA signaling in neurodegenerative diseases.
This review paper was well organized and described the almost all the issues of RhoA GTPase. This reviewer recommends to edit the manuscript with a little modification.
- P5, line 182 in GDF section, this reviewer recommends the description that IKKγ/NEMO can function as a GDF as it facilitates the dissociation between RhoA-RhoAGDI, resulting in RhoA activation (Ref: 155)
- P.7, first line in Table2, Tyr127 should be changed to Thr127.
Author Response
We thank for the positive review and have made the two suggested corrections:
1) IKKγ/NEMO is now added as a GDF for RhoA in line 178-179, page 5 and in Table 1.
2) Tyr127 in Table 2, page 7, has been changed to Thr127.
Reviewer 2 Report
The authors have written the review very well and have touched upon the signaling and how the role in diseases. The illustrations are done well which have added to the understanding to the review.
Author Response
We thank for the very positive review.
No changes has been made based on this review (no corrections suggested by the referee).
Reviewer 3 Report
The first part of this manuscript thoroughly describes the multiple regulation mechanisms, downstream signaling cascades, and functions of RhoA. In the second part, the authors summarized the involvement of RhoA in neurodegenerative diseases and discussed the therapeutic potential of the inhibition of RhoA signaling in those diseases, which is the main purpose of this article.
This review article will be useful for readers to obtain overall knowledge of RhoA, and shed light on the potential importance of RhoA in neurodegenerative diseases and their therapy.
Individual comments
Line 624; a -> α
Line 630; Not ROCK but ROCK-inhibition activates autophagy, doesn’t it?
Figures; Some labels and letters are illegible because of their colors and sizes (Figure 1A-C, Figure 2B, Figure 3), which could be improved.
Author Response
We thank for the positive review and find that the suggested changes have improved our mansucript. The following corrections have been made:
1) "a" had been replaced by "α", line 628 (previous line 624), page 20.
2) In line 634 (previous line 632), page 20, ROCK inhibition is described to activate autophagy.
3) The font size used for the figures (some letters) has been slightly enlarged/optimized, which we think improves their readability.
Reviewer 4 Report
In this manuscript “RhoA signaling in neurodegenerative diseases”, Schmidt at al., quite well explore the complex signaling pattern of RohA, Ras homolog gene family member A, involved in the regulation of several cellular functions, with a focus on both general functions and the involvement in neurodegenerative diseases such as Parkinson’s Disease, Alzheimer Disease, Huntington’s disease and amyotrophic lateral sclerosis (ALS).
The Authors well expose the complex signaling pathway of RohA. The complex analysis is well presented, from the structural domains to their functional roles, and it is well described with three clear and exhaustive cartoons, rich of information.
It is also supported by three highly informative Tables, which highlight RohA interacting factor, its post-translational modifications and the involvement, together with targeting factors, in the different models of neurodegeneration analysed, influenced by RohA signaling.
The Manuscript is generally well written, however some discussed arguments should be better addressed by essentially including more recent bibliographic references and the following specific comment to reach the required Journal’s standards, with only some Minor points:
- In the Introduction Section, line 60, the Authors could include a more general and short sentence about involvement of RohA signaling also in different forms of neurodegeneration, like traumatic brain injury or intellectual disability, to show a wider focus on the specific argument, over the principal neurodegenerations described;
- In the section “Post-transcriptional regulation of RohA by miRNAs”, the Authors well describe an important regulatory aspect by miRNAs,which could be highlighted with more recent bibliographic references to increase actuality, as well as in the 3.1 Section, “Cytoskeletal dynamics”, at line 404;
- In the general introduction of Section 3 “Downstream targets of RohA”, at line 376, the Authors should include more recent bibliography also related to neurodegeneration;
- In the complex signaling picture of RohA, the Authors well pointed out that its expression is both regulated by transcription factors, along with several reported microRNAs, and that RhoA itself is a transcriptional regulator with other factors. In the latter function, explained in the paragraph 3.6, the Authors should also include a note about the relevant involvement in cytoskeleton regulation by Sox9 and actin through RohA/ROCK signaling;
- In the sentence at line 569, section 3.6, the Authors could also mention the crosstalk of HIF-1 alpha with Rac 1 but not with RohA, in aged neurons, because of the important role of HIF-1 alpha in neuronal hypoxia ;
- In the Section “Parkinson’s Disease”, at the introduction level, lines 588-600, the support of wide and recent bibliography could be relevant for the argument.
Author Response
We thank the reviewer for the positive review and very useful suggestions for changes. We have modified the manuscript as follows:
Two recent references the role of RhoA/ROCK in traumatic brain injury, spinal cord injury and ischemic stroke have been included in the Introduction Section, and "traumatic brain injury" is now mentioned in the text (line 38, page 1).
A more recent reference on regulation of RhoA by miRNA has been provided in the revised manuscript (line 351, page 10). The reference in line 404 is commonly used - also in very recent literature (no recent alternative could be found). The section "Downstream target of RhoA"has been updated by two additional references (line 382, page 10).
Existing literature on SOX9 and RhoA mainly involve chondrocytes:
"Sox9 may play a role in RhoA mediated effects on chondrogenesis as RhoA-mediated modulation of actin-polymerization regulates Sox9 activity and Sox9 in turn positively regulates expression of chondrocyte markers" (Kumar & Lassar, 2009). Supported Kim et al., 2018. We are afraid that this information, based on chondrocytes, may confuse the message of our manuscript.
The Section "Parkinson's disease" (page 19) includes recent reviews - see ref. #216 and ref. #220. - both from 2020.
Reviewer 5 Report
In this work, Schmidt and colleagues reviewed the role of the RhoA pathway on neurodegenerative diseases. The manuscript is well structured, the information is well organized and covers most of the main aspects reported in the literature. I believe this manuscript can be published in its current form.
I have one last suggestion: can the authors discuss if external stimuli such electrical/magnetical/mechanical stimuli can impact the RhoA pathway as well? For example, recent literature evidenced a potential therapeutic role of electrical stimulation for neurodegenerative diseases. In some works, the authors found changes in the Akt pathway, which links to the RhoA/ROCK pathway, in the cells cytoskeleton and terminal differentiation into glutaminergic/GABAergic/dopaminergic neurons from NSCs.
Author Response
We thank the referee for the very positive review.
As suggested by the reviewer, we have been searching for literature linking electrical/magnetical and mechanical stimuli to Rho A signaling. Unfortunately, we were unable to identify studies and data of significant relevance or data that we feel will strengthen the impact or aim of our manuscript.